# Thiopurine S-Methyltransferase Polymorphisms Predict Hepatotoxicity in Azathioprine-Treated Patients with Autoimmune Diseases

**DOI:** 10.3390/jpm12091399

**Published:** 2022-08-28

**Authors:** Heh-Shiang Sheu, Yi-Ming Chen, Yi-Ju Liao, Chia-Yi Wei, Jun-Peng Chen, Hsueh-Ju Lin, Wei-Ting Hung, Wen-Nan Huang, Yi-Hsing Chen

**Affiliations:** 1Division of Allergy, Immunology and Rheumatology, Department of Internal Medicine, Taichung Veterans General Hospital, Taichung 40705, Taiwan; 2Department of Medical Research, Taichung Veterans General Hospital, Taichung 40705, Taiwan; 3School of Medicine, National Yang-Ming Chiao Tung University, Taipei 30010, Taiwan; 4Department of Post-Baccalaureate Medicine, College of Medicine, National Chung Hsing University, Taichung 40227, Taiwan; 5Rong Hsing Research Center for Translational Medicine & Ph.D. Program in Translational Medicine, National Chung Hsing University, Taichung 40227, Taiwan; 6Department of Pharmacy, Taichung Veterans General Hospital, Taichung 40705, Taiwan; 7Department of Medical Education, Taichung Veterans General Hospital, Taichung 40705, Taiwan; 8College of Business and Management, Ling Tung University, Taichung 408284, Taiwan

**Keywords:** TPMT genotype, TPMT poor metabolizers, TPMT intermediate metabolizers, AZA, hepatotoxicity, cumulative incidence of hepatotoxicity, pharmacogenomics, autoimmune disease, preemptive genotyping, TPMI

## Abstract

Thiopurine methyltransferase (TPMT) is the rate-limiting enzyme in Azathioprine (AZA) metabolization. Although studies have discussed the association between the TPMT polymorphisms and myelosuppression, the data about the relationship between TPMT genotypes and hepatoxicity in Asian patients remain limited. This study investigated the correlation between TPMT polymorphisms and AZA-related hepatotoxicity. This study enrolled the patients who had prior exposure to AZA from the Taichung Veterans General Hospital (TCVGH)-Taiwan Precision Medicine Initiative (TPMI) cohort. Genetic variants were determined using a single nucleotide polymorphism (SNP) array. Participants were accordingly categorized into normal metabolizer (NM) and non-normal metabolizer (non-NM) groups. From the TCVGH-TPMI cohort, we included 50 TPMT non-NM patients, including 1 poor metabolizer (PM), 49 intermediate metabolizers (IMs), and 1000 NM patients. The non-NM genotype was associated with hepatotoxicity compared with the NM genotype (hazard ratio (HR): 3.85, 95% confidence interval (CI): 1.83–8.10). In the non-NM group, the 3-year cumulative incidence of hepatotoxicity was higher than that in the NM group at 8.5% in the first year and 18.6% in the second and third years (*p* < 0.001). A TPMT non-NM genotype was associated with the occurrence of hepatotoxicity following AZA therapy. Preemptive testing helps individualize AZA therapy by minimizing the risk of hepatotoxicity.

## 1. Introduction

Azathioprine (AZA) is a valuable steroid-sparing immunosuppressant that is broadly used for systemic lupus erythematosus (SLE) [1,2,3], severe rheumatoid arthritis (RA), inflammatory bowel disease (IBD) [4,5,6,7], autoimmune hepatitis (AIH) [8], dermatomyositis (DM)/polymyositis, pemphigus [9], and post-transplant rejection. Azathioprine is a prodrug. Initially, AZA is metabolized to 6-mercaptopurine (6-MP) by glutathione S-transferase (GST). Three competitive enzymatic pathways metabolize 6-MP. First, xanthine oxidase (XO) catalyzes 6-MP to thiouric acid, an inactive metabolite. Second, thiopurine S-methyltransferase (TPMT), methylates 6-MP into 6-methylmercaptopurine (6-MMP). Third, hypoxanthine guanine phosphoribosyltransferase (HPRT) converts 6-MP into 6-thioinosine monophosphate (6-TIMP). Inosine monophosphate dehydrogenase (IMPDH) dehydrogenizes 6-TIMP into 6-thioxanthosine monophosphate (6-TXMP). Subsequently, guanosine monophosphate synthetase (GMPS) metabolizes 6-TXMP to 6-thioguanine nucleotides (6TGNs). 6TGNs, the primary active metabolites, integrate into DNA and RNA molecules to generate cytotoxic and therapeutic effects [10,11]. Patients with intermediate TPMT activity show 50% more 6-TGNs than those with normal or high TPMT activity. Low or absent TPMT activity can lead to further accumulation of 6-TGNs. Deficient enzyme activity may increase AZA-related side effects. AZA-related side effects include myelosuppression, which occurs in 3–17% of the patients [12,13], hepatotoxicity in up to 10% of patients [14], gastrointestinal adverse reactions (nausea, vomiting, and diarrhea), and hair loss.

Genetic polymorphisms of the TMPT gene have been studied extensively [15]. In the Caucasian population, approximately 4–11% of individuals have intermediate TPMT activity, while approximately 1 in 300 (0.3%) have very low or absent TPMT activity [16,17]. In the Asian population, the frequency of TPMT gene mutations is approximately 1.5–3%. TPMT polymorphisms were significantly associated with AZA-induced bone marrow toxicity [18,19,20]. Two meta-analyses by Liu et al. [19,20] reported the incidence of the TPMT polymorphisms was not associated with AZA-induced hepatotoxicity in patients with autoimmune diseases and IBD. However, the populations in these studies discussing AZA-induced hepatotoxicity were mainly Caucasian. The enrolled Asian population was limited. We may need a larger Asian population to evaluate the association between the incidence of AZA-induced hepatotoxicity and TPMT polymorphisms.

This retrospective case-control study aimed to evaluate whether there is a relationship between TPMT polymorphisms and the incidence of AZA-related hepatotoxicity.

## 2. Material and Methods

### 2.1. Study Design

This retrospective case-control study was performed in a single medical center in Taiwan with the approval of the Ethics Committee of Taichung Veterans General Hospital (SF19153A). Each participant offered written informed consent before study participation. This study was conducted according to the Declaration of Helsinki.

### 2.2. Study Population

The study participants were from the Taiwan Precision Medicine Initiative (TPMI), which collected information and specimens from Taiwanese volunteers from 15 hospitals throughout the nation. The tertiary referral medical center, Taichung Veterans General Hospital (TCVGH), contributed to the majority of the TPMI cohort. A total of 43,035 patients who were >20 years old were enrolled in the TPMI study from June 2019 to August 2021. A total of 2128 patients with present or previous exposure to AZA were selected. Patients who had not yet started therapy at the time of laboratory analysis were excluded. We excluded patients without a white blood cell (WBC) count, those without alanine transaminase (ALT) data, and those with a cancer diagnosis before taking AZA.

### 2.3. Data Collection

The date of AZA prescription was defined as index day. Clinical data after the index day, including age, gender, azathioprine dosage, first biochemistry profile, use of other immunosuppressive agents, comorbidities, and AZA prescription department were collected from the electric health record.

Comorbidities diagnosed once during hospitalization or at least twice in the outpatient department and six months before and after AZA prescription were included. These included hepatitis B (International classification of diseases (ICD) 10/9: B16, B18.0–B18.1, B19.10–B19.11/070.2–070.3), SLE (M32/710), Sjogren’s syndrome (M35/710.2), DM(M33.0–33.1, M33.9, M36.0/710.3), polymyositis (M33.2/710.4), pemphigus (L10.0–10.9/694.4), RA (M05.7–M05.9, M06.0, M06.2–M06.3, M06.8–M06.9, M08/714), systemic sclerosis (SSc) (M34/710.1), mixed connective tissue disease (M35.1/710.8), vasculitis (M30.0, M31.0, M31.3-M31.7/446.0, 446.2, 446.4–446.5, 446.7), pemphigoid (L12.1–12.9, K05.11/694.5–694.6), Behcet’s disease (M35.2/136.1), and IBD (K50, K51, K52.9/555.0–555.9, 556.0–556.9, 558.9).

Following the AZA therapy index day, the first onset time of leukopenia and hepatotoxicity was analyzed. Leukopenia was defined as a WBC count ≤ 4000/uL. Hepatotoxicity was defined as an ALT level ≥ 150 U/L. The AZA prescription division was also analyzed and included the divisions of immunology and rheumatology (IMRH), nephrology (NEPH), neurology (NEUR), dermatology (DERM), chest medicine (CM), cardiology (CV), gastroenterology and hepatology (GI), hematology (HEMA), metabolism, endocrinology and nutrition (META), gastrointestinal surgery (GS), Thoracic Surgery (CS), colorectal surgery (CRS), otolaryngology (ENT), ophthalmology (OPH), and pediatrics (PENP).

### 2.4. TMPT Genotyping

Genotyping was performed using the Taiwan Biobank (TWB, established in 2012) version 2 array [21], a single nucleotide polymorphism (SNP) array designed for the Taiwanese population by choosing optimized SNPs for imputation from the whole-genome sequencing (WGS) of the Taiwan Biobank participants. The TWBv2 array straightly genotypes more than 100,000 functional variants. The SNPs rs1142345 representing TPMT alleles *3A or *3C were analyzed. Alleles *3A and *3C were the common inactivating alleles in East Asia, accounting for 98% of all no-function alleles [22]. Genotyping tests including these variants are likely to be informative for the TPMT phenotype. A patient carrying one allele with the rs1142345 variant was reported as *1/*3 and was defined as intermediate metabolizer (IM). The patient carrying two alleles with the rs1142345 variant was reported as *3/*3 and was defined as a poor metabolizer (PM). According to genotyping, the patients were classified into TPMT normal metabolizers (NMs) and non-normal metabolizers (non-NMs), which included IMs and PMs.

### 2.5. Statistical Analysis

Data are presented as either numbers (percentages) or as the mean ± standard deviation (SD) or as the median (interquartile range). Comparisons were performed using the Mann–Whitney U test. Categorical data are presented as numbers and percentages and were compared using the chi-square test. Cox regression analysis was used to analyze which factors are related to hepatotoxicity with adjustment for covariates (TPMT genotype, age, gender, AZA dosage, other medication (methotrexate (MTX)), and comorbidity of hepatitis B). Kaplan–Meier curves were used to determine differences in cumulative incidence of hepatotoxicity between the TPMT non-NM and NM groups. A *p*-value ≤ 0.05 was considered statistically significant. All data were analyzed using the Statistical Package for the Social Sciences (SPSS, IBM Corp., Armonk, NY, USA) version 22.0.

## 3. Results

### 3.1. Study Patient Flow

Of the 43,035 patients enrolled in the TPMI, 2128 patients with AZA exposure were selected (Figure 1). A total of 628 patients without a WBC count, without ALT data, or with a cancer diagnosis before taking AZA were excluded. The TPMT non-NM and NM groups were matched by age and gender. There were 50 TPMT non-NM patients, including 49 IMs and 1 PM, and 1000 TPMT NM patients among the AZA users.

### 3.2. Demographics of the Selected Patients

The patient characteristics are summarized in Table 1. Patients were classified into TPMT non-normal metabolizer (non-NM) (*n* = 50) and normal metabolizer (NM) (*n* = 1000) groups. Age, gender, AZA dosage, post-AZA treatment first mean WBC count and ALT, and AZA prescription division were similar among the TPMT non-NM and NM groups. In terms of comorbidities, more patients had hepatitis B in the TPMT non-NM group than in the NM group.

### 3.3. TPMT Phenotypes and Risks of Hepatotoxicity and Leukopenia

The average dosage of AZA was not different between the TPMT non-NM and NM groups. However, more than half of the patients took less than 25 mg daily. A hepatotoxicity event was defined by an ALT level of more than 150 U/L, which showed no difference between the TPMT non-NM and NM groups. However, the onset time of hepatotoxicity was shorter in the TPMT non-NM group compared with the TPMT NM group (Table 2). Cox proportional hazards regression analysis was applied to evaluate the independent risk factors for hepatotoxicity following AZA treatment. Among 1050 AZA users, male gender (hazard ratio (HR): 1.77, 95% confidence interval (CI): 1.07–2.91), non-NM genotype (HR: 3.85, 95% CI: 1.83–8.10), and MTX use (HR: 1.62, 95% CI: 1.03–2.57) were associated with hepatotoxicity compared with the NM genotype. Age, AZA dosage, and hepatitis B carrier status were not associated with hepatotoxicity (Table 3). The cumulative incidence of hepatoxicity was significantly higher in the first three years in the TPMP non-NM group than in the NM group (*p* < 0.001). In the TPMP non-NM group, the 1-year cumulative incidence rate was 8.5%, the 2-year cumulative incidence rate was 18.6%, and the 3-year cumulative incidence rate was 18.6% (Figure 2).

## 4. Discussion

This was a hospital-wide retrospective case-control study. The patients were selected from the TPMI cohort in which the TPMT gene status was tested. We enrolled all AZA users. Among 1050 AZA users, the non-NM genotype, male gender, and MTX usage were independent risk factors for hepatotoxicity. The cumulative incidence of hepatoxicity was significantly higher in the first three years in the TPMT non-NM group than in the NM group. Our result shows that pre-emptive TPMT genotyping before starting AZA therapy may minimize the risk of hepatotoxicity.

Two meta-analyses by Liu et al. [19,20] reported that the incidence of TPMT polymorphisms was not associated with AZA-induced hepatotoxicity in patients with autoimmune diseases and IBD. Unlike the previous studies, our study revealed an association between TPMT polymorphisms and thiopurine-induced hepatotoxicity. First, our study design was different from those of the previous studies. We classified patients by TPMT genotyping before evaluating hepatotoxicity events. However, the earlier studies classified patients with hepatotoxicity events before assessing the TPMT genotype. Our study showed no difference in the first hepatotoxicity events between the TPMT non-NM and the NM groups. However, as the onset time of hepatotoxicity was considered and eliminating the interaction effect of variables by Cox regression analysis, the TPMT non-NM group was more prominent. Our study design was closer to the prospective concept. Second, our study population was different. Our patients had various autoimmune diseases, but the previous studies mainly analyzed patients with IBD. The subjects in the earlier studies discussing AZA-induced hepatotoxicity were primarily Caucasians. The Asian population was limited, but our study population was all Asians and was larger. As genetic testing is becoming less expensive and more accessible, pre-emptive TPMT testing is suggested in Asians to prevent post-AZA hepatotoxicity.

Previous studies revealed a significant association between TPMT polymorphisms and bone marrow toxicity. However, in our study, this was not noted. The previous studies primarily discussed Caucasians with IBD. The median dose of AZA in the previous studies was more than 1–2 mg/kg/day [23,24,25,26,27,28]. In contrast, our study enrolled patients with various autoimmune diseases, particularly SLE and SS. Our average AZA dosage was 36.4 mg in the TPMT non-NM group and 39.6 mg in the TPMT NM group. Approximately 90% of patients took less than 50 mg per day. We followed the AZA treatment strategy of starting with a lower dose and increasing the dose at a slow pace. Such a strategy seemed to keep the patients safe from myelosuppression even in those with the TPMT non-NM genotype. However, our study results show that taking AZA for a longer duration still has risks of hepatotoxicity.

In addition to the TPMT genotype, methotrexate usage and male gender were independent risk factors for hepatotoxicity. Clinical studies have widely reported the hepatotoxicity of MTX [29]. According to the FAERS database, females were less likely to develop liver injury than males [30]. A cross-sectional community study in Taiwan revealed that hepatitis with an elevated ALT was more common in men than in women [31]. According to our research, male patients with the TPMP non-NM genotype using MTX would have the highest risk of hepatotoxicity. These patients may take benefit from pre-emptive genotyping. This finding provides insights into the potential utilization of pharmacogenomics in individualized medical care.

There are several limitations to this study. First, the study design was retrospective. Missing data were inevitable. After three years of observation, the cumulative incidence of hepatotoxicity related to the TMPT non-NM genotype was noted. However, it is impossible to define a causal relationship between AZA and hepatotoxicity. The result was still significant. Second, we may have underestimated the number of cases of hepatotoxicity. There are three types of AZA-related hepatotoxicity: hepatocellular, cholestatic, and mixed. However, we only enrolled patients with hepatocellular-type hepatotoxicity. According to research by Siramolpiwat et al., the AZA-related hepatotoxicity type was predominantly a mixed type [32]. Evaluating only the ALT level may not miss patients with mixed-type hepatotoxicity. Thus, our hepatotoxicity event number was still reliable. Third, we only discussed the TMPT genotype. In addition to the TPMT genotype, our study did not discuss many other genetic polymorphisms, such as nucleoside diphosphate-liked moiety X motif 15 (NUDT15). NUDT15 was reported to be associated with more frequent adverse events than TPMT polymorphisms in the Asian population [33]. Further studies are needed to verify our results.

## 5. Conclusions

This hospital-wide retrospective case-control study supported the hypothesis that the TPMT non-NM genotype is associated with hepatotoxicity following AZA therapy. Pre-emptive testing for TPMT polymorphisms could help to individualize AZA therapy by minimizing the risk of hepatotoxicity, especially in male patients taking MTX.

## Figures and Tables

**Figure 1 jpm-12-01399-f001:**
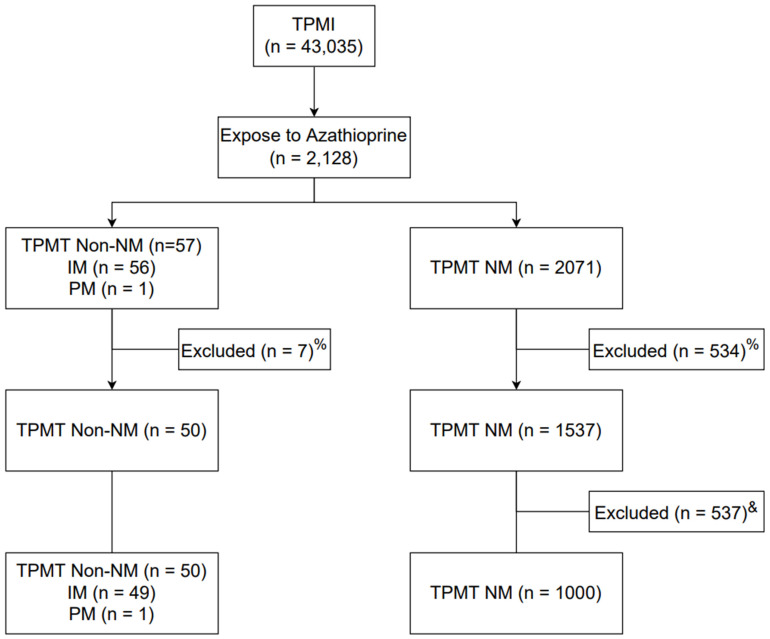
Patient enrollment flowchart. ^%^ 1. Exclude patients without WBC count and ALT data; 2. Exclude patients with a diagnosis of cancer before taking AZA. ^&^ Match by age and sex at 1:20. TPMI, Taiwan Precision Medicine Initiative; TPMT, thiopurine S-methyltransferase; NM, normal metabolizer; IM, intermediate metabolizer; PM, poor metabolizer; WBC, white blood cell; ALT, alanine transaminase; AZA, azathioprine.

**Figure 2 jpm-12-01399-f002:**
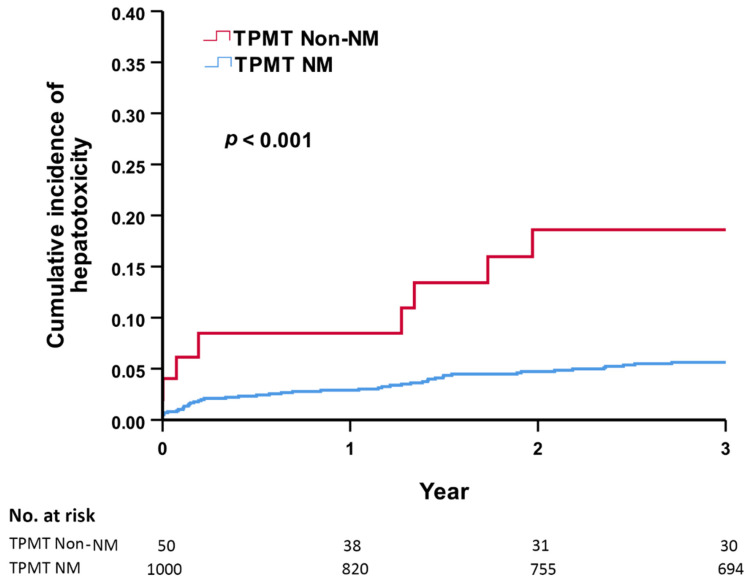
Three-year cumulative incidence of hepatotoxicity between the TMPT non-NM and NM groups. TPMT, thiopurine S-methyltransferase; NM, normal metabolizer.

**Table 1 jpm-12-01399-t001:** Clinical and demographic characteristics of patients in the TPMP non-NM and NM groups.

Variable	TPMT Non-NM (*n* = 50)	TPMT NM (*n* = 1000)	*p*-Value
Age	51.5 ± 11.7	51.4 ± 11.6	0.95
Gender			
Female	43 (86.0)	858 (85.8)	0.97
Male	7 (14.0)	142 (14.2)	
AZA dose (mg)	36.4 ± 17.9	39.6 ± 23.47	0.35
Biochemistry			
WBC count (U/L)	3920 (3600–5400)	3900 (3500–5660)	0.73
ALT (U/L)	13.5 (11–27)	13 (10–21)	0.16
Medication			
MTX	6 (12.0)	125 (12.5)	0.92
Cyclophosphamide	5 (10.0)	137 (13.7)	0.46
Comorbidity			
Hepatitis B	6 (12.0)	36 (3.6)	0.01
SLE	30 (60.0)	647 (64.7)	0.50
SS	24 (48.0)	456 (45.6)	0.74
DM and polymyositis	7 (14.0)	129 (12.9)	0.82
Pemphigus	3 (6.0)	25 (2.5)	0.14
Other autoimmune diseases ^	12 (24.0)	244 (24.4)	0.95
Division			
IMRH	42 (84)	864 (86.4)	0.52
NEPH	5 (10)	59 (5.9)	
NEUR	2 (4)	24 (2.4)	
DERM	0 (0)	27 (2.7)	
Others ^&^	1 (2)	26 (2.6)	

Data are expressed as *n* (%) or the mean ± SD or the median (interquartile range). ^ Other autoimmune diseases: RA, MCTD, vasculitis, SSc, Behcet’s disease, IBD. ^&^ Other divisions: CM, CV, GI, HEMA, META, GS, CS, CRS, ENT, OPH, PNEP. TPMT, thiopurine S-methyltransferase; NM, normal metabolizer; AZA, azathioprine; WBC, white blood cell; ALT, alanine transaminase; MTX, methotrexate; SLE, systemic lupus erythematosus; SS, Sjogren’s syndrome; DM, dermatomyositis; RA, rheumatoid arthritis; MCTD, mixed connective tissue disease; SSc, systemic sclerosis; IBD, inflammatory bowel disease; IMRH, immunology and rheumatology; NEPH, nephrology; NEUR, neurology; DERM, dermatology; CM, chest medicine; CV, cardiology; GI, gastroenterology and hepatology; HEMA, hematology; META, metabolism, endocrinology, and nutrition; GS, gastrointestinal surgery; CS, thoracic surgery; CRS, colorectal surgery; ENT, otolaryngology; OPH, ophthalmology; PENP, pediatrics.

**Table 2 jpm-12-01399-t002:** Outcome post-AZA treatment.

Variable	TPMT non-NM (*n* = 50)	TPMT NM (*n* = 1000)	*p*-Value
AZA dose (mg)			
≤25	27 (54.0)	563 (56.3)	0.21
25–50	21 (42.0)	333 (33.3)	
>50	2 (4.0)	104 (10.4)	
Outcome postAZA exposure			
Leukopenia ^ cases	28 (59.5)	611 (63.3)	0.60
Hepatitis ^&^ cases	10 (20.0)	150 (15.0)	0.33
Lowest WBC (U/L)	4375 (3210–5420)	4200 (3150–5390)	0.52
Highest ALT (U/L)	38 (22–84)	39 (26–73)	0.68
Onset of leukopenia ^ (days)	1359.3 ± 1709.7	1597.3 ± 1606.6	0.44
Onset of hepatitis ^&^ (days)	676.3 ± 837.9	2395.8 ± 1911.0	<0.0001

Data are expressed as *n* (%) or the mean ± SD or the median (interquartile range). ^ WBC count ≤ 4000 (U/L). ^&^ ALT level ≥ 150 (U/L). TPMT, thiopurine S-methyltransferase; NM, normal metabolizer; AZA, azathioprine; WBC, white blood cell; ALT, alanine transaminase.

**Table 3 jpm-12-01399-t003:** Cox regression analysis of risk factors for hepatotoxicity following AZA treatment.

	Hepatotoxicity ^		*p-*Value
	HR	95% CI	
Age	0.99	0.98–1.01	0.60
Gender (reference female)	1.77	1.07–2.91	0.03
TPMT non-NM genotype	3.85	1.83–8.10	0.0004
AZA dose (mg)	1.00	1.00–1.01	0.31
MTX	1.62	1.03–2.57	0.04
Hepatitis B carrier	1.18	0.40–3.48	0.76

^ Development of an ALT level ≥ 150 U/L after AZA treatment. ALT, alanine transaminase; TPMT, thiopurine S-methyltransferase; NM, normal metabolizer; AZA, azathioprine; MTX, methotrexate; HR: hazard ratio; CI: confidence interval.

## Data Availability

Not applicable.

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
