# Peer review of "Thiopurine S-Methyltransferase Polymorphisms Predict Hepatotoxicity in Azathioprine-Treated Patients with Autoimmune Diseases"

_jpm, 2022, doi:10.3390/jpm12091399_

Round 1
Reviewer 1 Report
The authors investigated the correlations between TPMT SNP polymorphism and the occurrence of hepatoxicity after Azathioprine exposure in an Asian group and found the non-NM genotype was associated with higher hepatotoxicity.
Major concern:
The major concern is the results in this study show inconsistency with regard to the correlation of TPMP genotype with hepatotoxicity. For instance, there are 10 patients with hepatotoxicity (ALT>150 IU/L) out of 50 non-NM patients, while 150 out of 1000 patients with NM genotype had hepatotoxicity. The correlation analysis showed no correlations (In Table. 2). Further, the ALT levels in hepatitis patients with non-NM genotype on onset days of leukopenia and hepatitis were much lower than those in patients with NM genotype (table. 2). The correlations between TPMT genotype and hepatotoxicity were only observed after cox regression analysis and in three-year cumulative incidence analysis. The contradicting results did not provide solid evidence to justify the correlations between TPMT polymorphism and hepatotoxicity.
Minor issues:
1. In figure 1, the asterisk was mistakenly labeled. Please correct it.
2. Are the ALT values in Table 1 correct? The mean values are less than 90 but the SD values are more than 140? The low levels of ALT values less than 150 IU/L also contradict the definition of hepatoxicity. Please also check the ALT values in Table. 2.
3. In Table. 2, the ALT values of patients with non-NM TPMT genotype on onset days of leukopenia and hepatitis were much lower than those of the NM patients. How to interpret the results?
4. Based on the y-axis and the data in Figure 2, I cannot understand why the cumulative incidences of hepatoxicity of patients with TPMT non-NM genotype were 8.5%, 18.6%, and 18.6%. Please indicate how did you calculate the cumulative incidence.
5. Please give more details on what the meanings represent in terms of the values shown in lines 99-103 on page 4.
6. The cited references in line 76 about Liu’s research were not correct. And in one of Liu’s meta-analyses, at least some previous studies from the Asian population were included. Therefore, the authors’ claim that there is no literature about the association between TPMT and hepatotoxicity in the Asian group is not correct.
7. Please give references in line 116 after the claim that alleles *3A and *3C represent 98% of all non-functional alleles.
Author Response
Point 1: Major concern: The major concern is the results in this study show inconsistency with regard to the correlation of TPMP genotype with hepatotoxicity. For instance, there are 10 patients with hepatotoxicity (ALT>150 IU/L) out of 50 non-NM patients, while 150 out of 1000 patients with NM genotype had hepatotoxicity. The correlation analysis showed no correlations (In Table. 2). Further, the ALT levels in hepatitis patients with non-NM genotype on onset days of leukopenia and hepatitis were much lower than those in patients with NM genotype (table. 2). The correlations between TPMT genotype and hepatotoxicity were only observed after cox regression analysis and in three-year cumulative incidence analysis. The contradicting results did not provide solid evidence to justify the correlations between TPMT polymorphism and hepatotoxicity.
Response 1: Thank you for the insightful question. The correlation analysis did not note the association between the TPMT genotype and hepatotoxicity. We considered it was due to the interaction effect between the variables. After eliminating the interaction effect with the Cox regression analysis, the association between TPMT genotype and hepatotoxicity was noted. (We revised the manuscript on page 12, line 14.)
The naming of items about outcome post azathioprine exposure in Table 2 may be misleading. We revised the naming of the items in Table 2. The value of the onset days of hepatitis in Table 2 represented the average onset days rather than the ALT level. The lower value meant that post-AZA exposure, the onset of the hepatotoxicity took a shorter time in the TPMT non-NM group compared with the TPMT NM group.
The result may not contradict the concept of the association between TPMT polymorphism and hepatotoxicity.
Point 2: Minor issue 1: In figure 1, the asterisk was mistakenly labeled. Please correct it.
Response 2: Thank you for the correction. The mistakenly labeled “asterisk” was corrected to “the percent sign(%)” in Figure 1.
Point 3: Minor issue 2: Are the ALT values in Table 1 correct? The mean values are less than 90 but the SD values are more than 140? The low levels of ALT values less than 150 IU/L also contradict the definition of hepatoxicity. Please also check the ALT values in Table. 2.
Response 3: Thank you for your insightful opinion. Because there are extreme values of the ALT and WBC, we changed our statistic method to the median and interquartile range. Therefore, we revised Table 1 and Table 2.
The ALT value in Table 1 included all the patients (with and without hepatotoxicity) in either TPMT non-NM or TPMT NM group, so the result may not contradict the definition of hepatotoxicity. According to the median of ALT in Table 1 and Table 2 after revision, the value of the defined hepatotoxicity would be above the third quartile. The hepatotoxicity cases were 20% in the TPMT non-NM group and 15% in the TPMT NM group. The results were compatible.
Point 4: Minor issue 3: In Table 2, the ALT values of patients with non-NM TPMT genotype on onset days of leukopenia and hepatitis were much lower than those of the NM patients. How to interpret the results?
Response 4: Thank you for your opinion. The naming of the items about outcome post azathioprine exposure in Table 2 may be misleading. We revised the naming of the items in Table 2. The “onset days of leukopenia and hepatitis” unit was “days.” So the lower value means hepatitis happened in a shorter period in the TPMT non-NM group compared with the TPMT NM group.
Point 5: Minor issue 4: Based on the y-axis and the data in Figure 2, I cannot understand why the cumulative incidences of hepatoxicity of patients with TPMT non-NM genotype were 8.5%, 18.6%, and 18.6%. Please indicate how did you calculate the cumulative incidence.
Response 5: Thank you for the insightful question. The definition of cumulative incidence is the number of new events or cases of a disease divided by the total number of individuals in the population at risk for a specific time interval. We calculated the cumulative incidence using the Statistical Package for the Social Sciences (SPSS, IBM Corp., Armonk, NY, USA) version 22.0, as shown in the table below.
Case |
Time |
Status |
Cumulative Proportion Surviving at the Time |
N of Cumulative Events |
N of Remaining Cases = N at risk |
The cumulative incidence |
||
Estimate |
Std. Error |
|||||||
1 |
0.000 |
1.00 |
0.980 |
0.020 |
1 |
49 |
|
|
2 |
0.000 |
0.00 |
|
|
1 |
48 |
|
|
3 |
0.003 |
1.00 |
0.960 |
0.028 |
2 |
47 |
|
|
4 |
0.038 |
0.00 |
|
|
2 |
46 |
|
|
5 |
0.074 |
1.00 |
0.939 |
0.034 |
3 |
45 |
|
|
6 |
0.077 |
0.00 |
|
|
3 |
44 |
|
|
7 |
0.077 |
0.00 |
|
|
3 |
43 |
|
|
8 |
0.140 |
0.00 |
|
|
3 |
42 |
|
|
9 |
0.142 |
0.00 |
|
|
3 |
41 |
|
|
10 |
0.192 |
0.00 |
|
|
3 |
40 |
|
|
11 |
0.192 |
1.00 |
0.915 |
0.041 |
4 |
39 |
0.0847 |
|
12 |
0.460 |
0.00 |
|
|
4 |
38 |
|
|
13 |
1.060 |
0.00 |
|
|
4 |
37 |
|
|
14 |
1.273 |
1.00 |
0.891 |
0.046 |
5 |
36 |
|
|
15 |
1.342 |
1.00 |
0.866 |
0.051 |
6 |
35 |
|
|
16 |
1.676 |
0.00 |
|
|
6 |
34 |
|
|
17 |
1.733 |
1.00 |
0.840 |
0.056 |
7 |
33 |
|
|
18 |
1.840 |
0.00 |
|
|
7 |
32 |
|
|
19 |
1.971 |
1.00 |
0.814 |
0.060 |
8 |
31 |
0.1859 |
|
20 |
2.979 |
0.00 |
|
|
8 |
30 |
|
|
21 |
3.091 |
0.00 |
|
|
8 |
29 |
|
|
22 |
3.173 |
0.00 |
|
|
8 |
28 |
|
|
23 |
3.671 |
0.00 |
|
|
8 |
27 |
|
|
24 |
3.685 |
0.00 |
|
|
8 |
26 |
|
|
25 |
3.704 |
0.00 |
|
|
8 |
25 |
|
|
26 |
3.795 |
0.00 |
|
|
8 |
24 |
|
|
27 |
3.981 |
0.00 |
|
|
8 |
23 |
|
|
28 |
4.194 |
0.00 |
|
|
8 |
22 |
|
|
29 |
5.706 |
0.00 |
|
|
8 |
21 |
|
|
30 |
5.736 |
0.00 |
|
|
8 |
20 |
|
|
31 |
5.760 |
0.00 |
|
|
8 |
19 |
|
|
32 |
5.774 |
0.00 |
|
|
8 |
18 |
|
|
33 |
5.788 |
0.00 |
|
|
8 |
17 |
|
|
34 |
5.791 |
0.00 |
|
|
8 |
16 |
|
|
35 |
5.936 |
0.00 |
|
|
8 |
15 |
|
|
36 |
6.174 |
0.00 |
|
|
8 |
14 |
|
|
37 |
6.779 |
0.00 |
|
|
8 |
13 |
|
|
38 |
7.458 |
0.00 |
|
|
8 |
12 |
|
|
39 |
7.551 |
0.00 |
|
|
8 |
11 |
|
|
40 |
8.961 |
0.00 |
|
|
8 |
10 |
|
|
41 |
9.087 |
0.00 |
|
|
8 |
9 |
|
|
42 |
9.933 |
0.00 |
|
|
8 |
8 |
|
|
43 |
10.023 |
0.00 |
|
|
8 |
7 |
|
|
44 |
10.494 |
0.00 |
|
|
8 |
6 |
|
|
45 |
11.069 |
0.00 |
|
|
8 |
5 |
|
|
46 |
13.112 |
0.00 |
|
|
8 |
4 |
|
|
47 |
13.810 |
0.00 |
|
|
8 |
3 |
|
|
48 |
18.415 |
0.00 |
|
|
8 |
2 |
|
|
49 |
19.937 |
0.00 |
|
|
8 |
1 |
|
|
50 |
20.057 |
0.00 |
|
|
8 |
0 |
|
Point 6: Minor issue 5: Please give more details on what the meanings represent in terms of the values shown in lines 99-103 on page 4.
Response 6: Thank you for the reminder. The values meant the international classification of diseases (ICD) codes of the tenth and ninth revisions. The values before a forward slash are the ICD-10 codes of the related diseases, and the values after a forward slash are the ICD-9 codes. The manuscript was revised on page 4, line 19 & 23.
Point 7: Minor issue 6: The cited references in line 76 about Liu’s research were not correct. And in one of Liu’s meta-analyses, at least some previous studies from the Asian population were included. Therefore, the authors’ claim that there is no literature about the association between TPMT and hepatotoxicity in the Asian group is not correct.
Response 7: Thank you for the correction. The cited references in line 76 about Liu’s research were corrected. The manuscript was revised on page 3, line 21. In both, the meta-analyses by Liu et al., the population of AZA-induced hepatotoxicity was mainly Caucasian. The Asain population was limited. Only one Asian population-based study related to AZA-induced hepatotoxicity was included in each meta-analyze. Zhu Q et al. study included 52 Chinese IBD patients treated with AZA with only one hepatotoxicity. [Thiopurine methyltransferase gene polymorphisms and activity in Chinese patients with inflammatory bowel disease treated with azathioprine. Chin Med J (Engl). 2012 Oct;125(20):3665-70.] Chen D et al. included 126 Chinese SLE patients treated with AZA with only three hepatotoxicity cases. [Association of thiopurine methyltransferase status with azathioprine side effects in Chinese patients with systemic lupus erythematosus. Clin Rheumatol. 2014 Apr;33(4):499-503.] The manuscript was revised on page 3, line 23 and on page 12, line 17.
Point 8: Minor issue 7: Please give references in line 116 after the claim that alleles *3A and *3C represent 98% of all non-functional alleles.
Response 8: Thank you for the question. The reference [Clinical Pharmacogenetics Implementation Consortium Guideline for Thiopurine Dosing Based on TPMT and NUDT15 Genotypes: 2018 Update. Clin Pharmacol Ther. 2019 May;105(5):1095-1105.] was added. The manuscript was revised on page 5, line 11.
According to the TPMT allele functionality and frequency tables included in the supplement, as shown in the table below, TPMT*3C is the most common no-function allele in the East Asian population, followed by TPMT*3A. The result of TPMT*3A and *3C represented 98% of all non-functional alleles came from the formula below.
(TPMT*3A+TPMT*3C) allele frequency/All no function allele frequency
=0.0164+0.0003 / 0.0170849
=0.977
TPMT allele |
Frequencies of TPMT alleles in East Asian |
Allele clinical functional status |
*1 |
0.9796 |
Normal Function |
*2 |
0.0000849 |
No Function |
*3A |
0.0003 |
No Function |
*3B |
0 |
No Function |
*3C |
0.0164 |
No Function |
*4 |
0 |
No Function |
*5 |
0 |
Uncertain Function |
*6 |
0.0022 |
Uncertain Function |
*7 |
0 |
Uncertain Function |
*8 |
0 |
Uncertain Function |
*9 |
0 |
Uncertain Function |
*10 |
Uncertain Function |
|
*11 |
No Function |
|
*12 |
0.0000531 |
Uncertain Function |
*13 |
Uncertain Function |
|
*14 |
0 |
No Function |
*15 |
0 |
No Function |
*16 |
0.0002 |
Uncertain Function |
*17 |
Uncertain Function |
|
*18 |
0 |
Uncertain Function |
*19 |
Uncertain Function |
|
*20 |
0 |
Uncertain Function |
*21 |
0 |
Uncertain Function |
*22 |
Uncertain Function |
|
*23 |
0 |
No Function |
*24 |
0 |
Uncertain Function |
*25 |
Uncertain Function |
|
*26 |
0.0003 |
Uncertain Function |
*27 |
Uncertain Function |
|
*28 |
Uncertain Function |
|
*29 |
0.0003 |
No Function |
*30 |
0 |
Unknown Function |
*31 |
0 |
Uncertain Function |
*32 |
0 |
Uncertain Function |
*33 |
0 |
Uncertain Function |
*34 |
0 |
Uncertain Function |
*35 |
Unknown Function |
|
*36 |
Unknown Function |
|
*37 |
0 |
Uncertain Function |
*38 |
0.0004 |
Unknown Function |
*39 |
0.0001 |
Uncertain Function |
*40 |
0 |
Uncertain Function |
*41 |
No Function |

Reviewer 2 Report
This is a well structured article regarding TPMT polymorphisms and occurrence of hepatotoxicity following AZA therapy. Although the data reported are solid, there are a few minor points that should be addressed:
1) The authors should include a statistical power analysis of their study.
2) How confident are the authors in studying the variants of only one gene in predicting hepatotoxicity due to AZA therapy?
3) I suggest giving some more information regarding "Taiwan Biobank version 2 (TWBv2) array".
Author Response
Point 1: The authors should include a statistical power analysis of their study.
Response 1: Thank you for the recommendation. A two-sided log-rank test was done for the statistical power analysis of our research. It achieved 81.5% power at a 0.050% significance level.
Point 2: How confident are the authors in studying the variants of only one gene in predicting hepatotoxicity due to AZA therapy?
Response 2: Thank you for the question. We would discuss the question from two perspectives.
Regarding the TPMT gene, we only analyzed the SNPs rs1142345. Because TPMT*3C was represented by the SNPs rs1142345, and the SNPs rs1142345 and rs1800460 represented TPMT*3A. So Analyzing the SNPs rs1142345 could get the phenotype. TPMT*3C is the most common no-function allele in the East Asian population, followed by TPMT*3A. According to the TPMT allele functionality and frequency tables included in the Clinical Pharmacogenetics Implementation Consortium Guideline supplement, the result of TPMT*3A and *3C represented 98% of all non-functional alleles.
Besides the TPMT gene, the NUDT15 gene was reported to be associated with more frequent adverse events in the Asian population. But the adverse event mainly discussed bone marrow suppression. We may need further study to evaluate the correlation between NUDT15 polymorphisms and hepatotoxicity. We mentioned it in the limitation of our study (on page 13, line 15).
Point 3: I suggest giving some more information regarding "Taiwan Biobank version 2 (TWBv2) array".
Response 3: Thank you for the suggestion. The TWBv2 array utilized whole-genome sequencing (WGS) data from TWB participants to choose SNPs optimized for imputation in Han Chinese samples, contained 114,000 risk variants in 2831 rare disease genes selected from published literature and the ClinVar database, 4100 variants associated with drug metabolism and adverse drug reactions, and 24,865 copy number variation (CNV) probes corresponding to known chromosomal aberrations and CNV regions.
By design, the TWBv2 array directly genotypes more than 100,000 functional variants, including mutations causing Mendelian diseases, variants associated with complex disease susceptibility, mutations known to affect drug metabolism, and variants across the HLA region. After annotation and sequence validation, we tabulated the genotype frequencies of disease-causing or pathogenic risk variants with minor allele frequency (MAF) > 0.1% in the TWB participants typed on the TWBv2 array. (We revised the material and methods section on page 5, line 7.)
